# Genomics of Re-Emergent *Aeromonas salmonicida* in Atlantic Salmon Outbreaks

**DOI:** 10.3390/microorganisms12010064

**Published:** 2023-12-29

**Authors:** Marcos Godoy, Marco Montes de Oca, Rudy Suarez, Alexis Martinez, Juan Pablo Pontigo, Diego Caro, Karina Kusch, Yoandy Coca, Harry Bohle, Sion Bayliss, Molly Kibenge, Frederick Kibenge

**Affiliations:** 1Centro de Investigaciones Biológicas Aplicadas (CIBA), Puerto Montt 5501842, Chile; marco.montesdeoca@ciba.cl (M.M.d.O.); diego.caro@ciba.cl (D.C.); karina.kusch@ciba.cl (K.K.); 2Laboratorio de Biotecnología Aplicada, Facultad de Ciencias de la Naturaleza, Escuela de Medicina Veterinaria, Universidad San Sebastián, Sede de la Patagonia, Puerto Montt 5480000, Chile; 3Programa de Magíster en Acuicultura, Facultad de Ciencias del Mar, Universidad Católica del Norte, Coquimbo 1780000, Chile; rudysuarez.vet@gmail.com; 4ATC Patagonia S/N, Carretera Austral, Puerto Montt 5480000, Chile; alexismartinez@atcpatagonia.com; 5Laboratorio Institucional, Facultad de Ciencias de la Naturaleza, Medicina Veterinaria, Universidad San Sebastián, Lago Panguipulli 1390, Puerto Montt 5501842, Chile; juan.pontigo@uss.cl; 6Doctorado en Ciencias de la Ingeniería, Departamento de Ingeniería Química y Bioprocesos, Escuela de Ingeniería, Pontificia Universidad Católica de Chile, Avenida Vicuña Mackenna 4860, Santiago 7820436, Chile; ycoca@uc.cl; 7Laboratorio InnovoGen, Egaña 198 Piso 2, Puerto Montt 5502534, Chile; hbohle@grupoinnovo.cl; 8Bristol Veterinary School, University of Bristol, Bristol BS8 1QU, UK; s.bayliss@bristol.ac.uk; 9Department of Pathology and Microbiology, Atlantic Veterinary College, University of Prince Edward Island, 550 University Ave, Charlottetown, PE C1A 4P3, Canada; mkibenge@upei.ca (M.K.); kibenge@upei.ca (F.K.)

**Keywords:** furunculosis, *Aeromonas salmonicida*, *Salmo salar*, genomic diversity

## Abstract

Furunculosis, caused by *Aeromonas salmonicida*, poses a significant threat to both salmonid and non-salmonid fish in diverse aquatic environments. This study explores the genomic intricacies of re-emergent *A. salmonicida* outbreaks in Atlantic salmon (*Salmo salar*). Previous clinical cases have exhibited pathological characteristics, such as periorbital hemorrhages and gastrointestinal abnormalities. Genomic sequencing of three Chilean isolates (ASA04, ASA05, and CIBA_5017) and 25 previously described genomes determined the pan-genome, phylogenomics, insertion sequences, and restriction-modification systems. Unique gene families have contributed to an improved understanding of the psychrophilic and mesophilic clades, while phylogenomic analysis has been used to identify mesophilic and psychrophilic strains, thereby further differentiating between typical and atypical psychrophilic isolates. Diverse insertion sequences and restriction-modification patterns have highlighted genomic structural differences, and virulence factor predictions can emphasize exotoxin disparities, especially between psychrophilic and mesophilic strains. Thus, a novel plasmid was characterized which emphasized the role of plasmids in virulence and antibiotic resistance. The analysis of antibiotic resistance factors revealed resistance against various drug classes in Chilean strains. Overall, this study elucidates the genomic dynamics of re-emergent *A. salmonicida* and provides novel insights into their virulence, antibiotic resistance, and population structure.

## 1. Introduction

Furunculosis is a bacterial disease that is caused by *Aeromonas salmonicida*, which affects various species of salmonid and non-salmonid fish in both freshwater and marine environments and is characterized by high mortality and morbidity. *A. salmonicida* is a Gram-negative, facultatively anaerobic, and nonmotile bacterium which is classified within the genus *Aeromonas* and belongs to the class Gammaproteobacteria and the family Aeromonadaceae [1]. The bacterium exhibits a rod-shaped morphology, measuring approximately 1.3–2.0 by 0.8–1.3 μm. Moreover, it possesses the ability to both ferment and oxidize glucose and exhibits positive results in catalase and cytochrome oxidase tests. It particularly affects salmonids and trout, thereby making it a significant concern in aquaculture. *A. salmonicida* can be divided into five subspecies: *salmonicida*, *smithia*, *achromogenes*, *masoucida*, and *pectinolytica* [2,3,4,5,6,7,8,9]. The *A. salmonicida* subspecies *salmonicida* represents the classical causative agent of furunculosis in salmonids, although atypical strains of *A. salmonicida* such as *masoucida*, *achromogenes*, and *smithia* can cause a variant in non-salmonids and warm-water fish known as atypical furunculosis [10,11,12]. Notably, among the known subspecies of *A. salmonicida*, *A. salmonicida pectinolytica* has been observed growing at temperatures exceeding 35 °C [9], while the other subspecies are restricted to lower temperatures. Further, several mesophilic strains have been described, for instance strain A527 initially recovered at 30 °C, which grows efficiently at both 18 °C and 37 °C [13].

Atypical furunculosis induces systemic bacterial disease in salmonids. It manifests as rapid and minimally symptomatic peracute cases, resulting in characteristic lesions, and in acute furunculosis, leading to high mortality and widespread tissue damage. The infection affects key organs, including the kidney, gills, spleen, myocardium, liver, and other internal structures, causing hemorrhages and necrosis. Externally, the disease is characterized by the development of skin ulcers and muscle cavities, among other symptoms. Internally, atypical furunculosis involves perirenal hemorrhages and hemorrhages in visceral fat. Histologically, bacterial colonies can be observed in various organs such as the liver, kidneys, and heart, among others [3,14,15]. The first report of atypical furunculosis in Atlantic salmon (*S. salar*) cultured in Chilean seawater was described in 2000 [16]. Subsequently, atypical furunculosis spread rapidly in estuarine, marine, and freshwater salmon farms [16,17]. In 2001, an injectable intraperitoneal vaccination containing inactivated atypical *A. salmonicida* was initiated in Atlantic salmon (*S. salar*), which reduced disease prevalence [18]. Of the total mortality due to infectious causes in the freshwater phase of Atlantic Salmon (*S. salar*), the incidence of atypical furunculosis (atypical *A. salmonicida*) increased from less than 0.1% reported in the years 2020 and 2021 to 2.5% reported in the year 2022 [19]. Therefore, it is considered a re-emerging infection, indicating a resurgence or new outbreaks of old infectious diseases, with significant implications [20].

Recently, various aspects contributing to the genomic diversity of *A. salmonicida* have been described, for instance the identification of distinct genomic islands (GEIs). For several *A. salmonicida* subp. *salmonicida* isolates, the GEIs *AsaGEI1a*, *AsaGEI1b*, *AsaGEI2a*, *AsaGEI2b*, *AsaGEI2c*, and *AsaGEI2d* have been described [21,22,23,24]. These have revealed noteworthy geographical associations. Therefore, the identification of GEIs might be useful as geographical origin indicators.

Moreover, the establishment and implementation of a *vapA*-centric typing system facilitated the subtyping of 675 isolates of *A. salmonicida*. This system relies on sequence variations within the virulence array protein (vapA) gene, which encodes the A-layer surface protein array in *A. salmonicida*. This innovative approach led to the identification of nine novel subtypes (15–23), with certain subtypes exhibiting a discernible affinity for specific fish species, notably, Cyprinidae and Pleuronectidae [25]. Furthermore, the sequencing and comparative analyses of entire genomes from *A. salmonicida* strains provided further insights into the genomic distinctions between typical and atypical strains. The genomic disparities were scrutinized by evaluating the insertion sequences (ISs) [26], along with identifying variations in the presence and absence of genes associated with virulence factors, transcriptional regulators, and non-coding RNAs. Additionally, a discernible role of the plasmidome in both the virulence and genomic adaptability of *A. salmonicida* has been highlighted [27]. 

Similarly, a comparative examination of 25 complete *A. salmonicida* genomes revealed gene clusters exclusive to the psychrophilic clades, which are associated with the lateral flagella, outer membrane proteins (including A-layer and T2SS proteins), and insertion sequences (specifically IS*As4*, IS*As7*, and IS*As29*). In contrast, the mesophilic clade exhibited the exclusive presence of MSH type IV pili, which suggests that discernible genetic factors are linked to the specific lifestyle of *A. salmonicida* within this particular clade [28]. 

Furthermore, the comprehensive analysis of extensive genomic data compiled from hundreds of *Aeromonas* strains in aquaculture has enabled the detailed profiling of resistome patterns and can offer insights into associations by isolation year, country of origin, and species characteristics. Indeed, over 100 antibiotic resistance genes (ARGs) have been identified within a dataset encompassing approximately 400 *A. salmonicida*, *A. hydrophila*, *A. veronii*, *A. media*, and *A. sobria* genomes, which display either species-specific or non-species-specific patterns. Notably, species such as *A. salmonicida* and *A. media* exhibit higher proportions of species-specific ARGs, indicating distinctive patterns in the acquisition of antibiotic resistance. The prevalence of genes such as *sul1*, *tet(A)*, and *tet(D)* has also been determined in specific regions of Asia and North America. This underscores the importance of regulating and controlling the use of antibiotics in aquaculture to mitigate the escalation of antibiotic resistance [29].

Prokaryotic defense mechanisms against foreign genomes involve crucial components known as restriction-modification (R-M) systems, which are present in a diverse range of unicellular organisms, encompassing both eubacteria and archaea [30,31]. These systems include two distinct enzymatic functions: restriction endonuclease (REase) and methyltransferase (MTase) functions. The REase targets and cleaves foreign DNA sequences at specific locations, while MTase activity maintains a distinction between the DNA of the host and any external DNA by adding methyl groups to the same specified DNA sequences. R-M systems exhibit diversity in their classifications, with distinctions based on cleavage position, co-factor requirements, sequence recognition, subunit composition, and substrate specificity, thus resulting in four primary types [32]. 

However, R-M systems seem to play multifaceted roles in bacterial biology, and their functions extend beyond merely protecting against foreign genetic material and include modulating horizontal gene transfer (HGT) and differentiating among bacterial strains. Due to the rampant impact of HGT on prokaryotic species boundaries, bacterial species are sustained through genetic isolation. REases operate as regulators by restricting foreign DNA with non-native methylation patterns to control the genetic influx. This creates a barrier, which could also serve to preserve the bacterial species [33]. 

Although atypical furunculosis has been historically reported among Atlantic salmon (*S. salar*) in Chile, remarkably a draft genome of *A. salmonicida* isolated from infected rainbow trout (*Oncorhynchus mykiss*) in Chile has been reported previously [34]. To the best of our knowledge, closed genomes of *A. salmonicida* isolated in Chile from cultivated Atlantic salmon (*S. salar*) are not currently available. In this study, we describe re-emergent *A. salmonicida* outbreaks among Atlantic salmon (*S. salar*) in Chile. We sequenced and closed genomes from three strains isolated from the X region. We characterized their virulence and antibiotic resistance factors, as well as plasmid content. Further, by comparing these genomes with 25 closed genomes available on NCBI [35], we corroborate recent findings regarding genomic variation patterns among typical psychrophilic, atypical psychrophilic, and mesophilic strains of *A. salmonicida* [26,27,28]. Finally, we describe the R-M landscape among *A. salmonicida* strains.

## 2. Materials and Methods

### 2.1. Sample Collection and Gross Pathology

The investigated fish were obtained from different hatcheries located in the regions of Los Lagos and Magallanes at the pre-smolt and smolt stages. Fish samples were collected at each hatchery by selecting the tanks with high mortality rates due to the furunculosis outbreak. Fish exhibiting at least one of the following symptoms were sampled: erratic or surface swimming, exophthalmia, mycosis, fin base hemorrhages, or skin lesions. Necropsies were performed to confirm the diagnosis and establish pathological characteristics in each case, following the description provided by [36]. Additionally, samples were taken from internal organs (kidney and spleen) for molecular biology analysis (PCR for atypical *A. salmonicida*), internal organs fixed in formalin were selected for histopathological analysis, and fresh fish samples were sent to the laboratory for culturing and to isolate atypical *A. salmonicida* onto solid media (Tryptic Soy Agar). For cases where successful bacterial isolation was achieved, genomic DNA (gDNA) was directly extracted from colonies, and whole bacterial genome sequencing (WGS) was performed.

### 2.2. Histopathology

Tissue samples intended for histological analysis were gathered and preserved in 10% buffered formalin. Following collection, they underwent standard processing procedures. The resulting sections, measuring 3–4 µm in thickness, were subsequently stained with hematoxylin and eosin (H&E) as per the standard protocol [37] to identify the significant microscopic morphological changes.

### 2.3. DNA Extraction and PCR

Here, automated tissue homogenization was conducted on the samples using the MagNA Lyser instrument (Roche, Basel, Switzerland). Subsequently, a robot was employed to perform total DNA extraction (Roche MagNA Pure LC instrument, Basel, Switzerland) alongside the MagNA Pure LC RNA isolation kit III (Tissue) following the manufacturer’s instructions. The extracted DNA was eluted in 50 µL of nuclease-free water, and DNA yields were quantified, while the DNA purity was analyzed by using a NanoPhotometer^®^ P 300 (Implen, Westlake Village, CA, USA) using the OD_260/280_ ratio. Following quantitation, the eluted DNA was promptly tested for the detection of *A. salmonicida*, using thermocycling conditions and specific primers for the surface array protein *VapA* gene, as previously described [38]. Piscine orthoreovirus (PRV), infectious salmon anemia virus (ISAV), and infectious pancreatic necrosis virus (IPNV) detection was also performed as described previously [39,40,41].

### 2.4. Bacterial Isolation

The main organs selected for bacterial isolation were the spleen, kidney, liver, and skin lesions, as previously described [42,43]. The selected organs were aseptically streaked onto tryptic soy agar (TSA) plates and incubated at 16 °C for 5 days. The purification process was applied to bacteria exhibiting morphological traits indicative of *A. salmonicida*. Colonies meeting purity and displaying consistent characteristics described in the existing literature—distinct white-to-gray color, well defined with irregular edges, and devoid of brown pigment production [42,43] —underwent microscopy (Gram stain) and PCR [38]. These analyses aimed to confirm the bacterial species as atypical *A. salmonicida*. gDNA was extracted using the Total DNA kit (Omega-Biotek, Norcross, GA, USA), following the manufacturer’s instructions. The purified gDNA concentration was determined using a NanoDrop spectrophotometer. In total, 100 ng of gDNA from each strain was utilized for subsequent library preparations.

### 2.5. Whole Genome Sequencing (WGS) and Bioinformatic Analysis

The ASA04 and ASA05 strains were sequenced using NanoPore technology. The extracted gDNA was prepared for sequencing using the Rapid Sequencing Kit V14, following the manufacturer’s guidelines: This kit is specifically optimized for the MinION Mk1C sequencer (Oxford Nanopore Technologies, Oxford, UK), thereby ensuring efficient library preparation and optimal sequencing performance. The prepared libraries were loaded onto a FlowCell R10.4.1 and sequenced by using the MinION Mk1C sequencer. The device was calibrated for dual-read sequencing to enhance accuracy by reading both DNA strands. After sequencing, the raw electric signal data from the MinION Mk1C were converted into nucleotide sequences through the Guppy basecalling software (v6.5, CUDA v10). Default parameters were employed, as recommended by the manufacturer. Then, the basecalled reads obtained from Guppy were utilized for genome assembly. Two separate assemblers, Canu (v2.2) [44] and Flye (v2.9.2) [45], were engaged to ensure that the assembly was accurate and robust. The achieved coverage values for the assembled genomes were 99.73× and 99.04×, respectively.

The CIBA_5017 strain was sequenced by PacBio Technology. The extracted gDNA was prepared for sequencing using the SMRTbell prep kit 3.0. The prepared library was sequenced using the Sequel II sequencer. Using subreads as input, circular consensus sequences (CCS) were generated with a minimum of 3 passes and a minimum predicted accuracy of 0.99 by using the CCS (v.4.2.0) tool available in SMRT Link v9.0. This allows for the resulting CCS reads to have a high accuracy of >99% and Q > 20. The resulting CCS reads were used as input for downstream analysis. 

The FASTQ file reads from the three strains were checked for quality using fastp (v0.23.2) [46], and raw read statistics are summarized in Appendix A. The PacBio reads were assembled with Unicycler (v0.5.0) [47], achieving a coverage value of 121x. The quality of the assemblies was checked using the QUAST tool (v5.0.2) [48]. Genome annotation and pan-genome analysis were performed using Prokka (v1.13) [49] and PIRATE (v1.0.5) [50], respectively. A core genome assembly encompassing SNPs was built with Snippy (v4.6.0) [51], and from it, a maximum likelihood phylogenetic tree was built using W-IQ-TREE (v1.6.12), the web interface and server for IQ-TREE [52,53,54,55]. This tree was visualized using iTOL (v6.8.1) [56]. Virulence factors, antibiotic resistance factors, insertion sequences, and R-M system enzymes were predicted using the VFDB [57], CARD [58], ISFinder [59], and REBASE [60] databases, respectively, through local alignments with BLAST (v2.12.0+) [61]. Plasmid composition was visualized with the Proksee tool CGView Builder (v1.1.4) [62]. Average nucleotide identity (ANI) values were computed with FastANI (v1.33) [63]. From the orthologous gene families predicted with PIRATE (v1.0.5) [50], the percentages of orthologous common proteins (POCPs) were computed with respect to the total number of gene families of the strain where this number was greater for each pairwise comparison. Heatmaps and clustermaps were processed and visualized with the Python libraries pandas (v1.3.4) [64], matplotlib (v3.4.3) [65], and seaborn (v0.11.2) [66]. Pangenome output was visualized as a flower plot with the flower-plot script (v0) [67].

## 3. Results

### 3.1. Laboratory Analysis of Clinical Cases

Overall, 27 clinical cases of atypical furunculosis were reported among Atlantic salmon (*S. salar*) during 2022 in Chile and analyzed in this study. The epidemiological background and laboratory results are summarized in Table 1. These showed an average mortality rate of 4.66% and an outbreak time of 8 weeks. Three bacterial isolates obtained from clinical cases were selected for genome sequencing. All the isolates were from clinically diseased fish, specifically those exhibiting the highest frequency in the clinical findings. None of the isolated strains produced pigmentation.

### 3.2. Gross Pathology

Some of the clinical signs illustrated were decreased appetites, increased lethargy, and exophthalmia (protrusion of the eyeball). Externally, the affected fish may have exhibited periorbital hemorrhages (around the eyes), hemorrhages at the base of the fins, and hemorrhages in the abdomen, anus, and the peduncle. Sometimes, the presence of bullae, which are fluctuating areas upon palpation called furuncles which correspond to muscular necrosis, could also be observed. Further, desquamation, erosion, and/or skin ulcers, whether singular or multiple, individual or coalescent, resulted in exposure of the musculature and edges displaying a white or red coloration. Internally, the affected fish exhibited petechial hemorrhages in the liver, swim bladder, heart, visceral fat, and perirenal hemorrhages; variable findings may include nephromegaly (enlarged kidneys) and splenomegaly (enlarged spleen). Finally, hemorrhagic enteritis and hyperemic gastric mucosa were observed in the gastrointestinal system (Figure 1 and Figure 2).

### 3.3. Histopathology

Variable colonies of bacteria were found to be replacing the tissue in the affected fish, with limited inflammatory responses. All organs were affected by the bacterial colonies, with the musculature, heart, gills, spleen, kidney, and liver being the most frequently affected tissues. In the skin, necrosis of the epidermis and dermis was observed, which deeply affected the adjacent musculature, with necrosis of the muscle fibers and associated hemorrhages, along with an abundance of bacterial colonies. There was hemorrhagic necrosis in the affected muscles, which led to the formation of cavities containing bloody fluids, sometimes with a viscous consistency (Figure 3). Additionally, the presence of hyperplasia and nephropathy was detected in a variable manner, which was possibly associated with suboptimal water quality parameters. Myocarditis, myositis, and hepatic necrosis were also observed, although these findings can be explained by infections from PRV, the causal agent of skeletal and cardiac muscle disease (HSMI) in Atlantic salmon (*S. salar*). Furthermore, peritonitis findings were identified, which are potentially linked to secondary reactions resulting from the intraperitoneal vaccination.

### 3.4. Pan-Genome Characterization

The PIRATE toolbox was applied to 28 complete *A. salmonicida* genomes, from which 3 strains are reported in this study and 25 were retrieved from NCBI [35] as described in a previous pangenome characterization [28]. The pangenome of *A. salmonicida* comprised 7296 gene families, of which 3364 (46.11%) were classified as core (>95% genomes) and 3932 (53.89%) as accessory. The analysis of the present/absent gene families showed remarkably distinct patterns between the mesophilic and psychrophilic strains, as well as within the typical and atypical psychrophilic strains. Moreover, the landscape of the present gene families seems to be heterogeneous among the atypical psychrophilic strains (Figure 4A). The genomes described in this study, ASA04, ASA05, and CIBA_5017, contained 17, 2, and 2 unique gene families, respectively (Figure 4B).

### 3.5. Phylogenomics

The obtained maximum likelihood dendrogram shows the grouping of the 28 analyzed *A. salmonicida* genomes into two main clusters (Figure 5). One is composed of mesophilic strains (red in Figure 5) and is distantly located toward the second cluster (composed of psychrophilic strains). Interestingly, the psychrophilic group is further divided into two subclusters; one is composed of typical psychrophilic strains (green in Figure 5), while the second is composed of atypical psychrophilic strains (yellow in Figure 5). The three strains described in this study cluster together within the atypical psychrophilic group. Notably, the mesophilic group shows the higher genomic divergence among *A. salmonicida* strains. This is also reflected by average nucleotide identity (ANI) values (Appendix A) and the percentage of orthologous common proteins (POCPs) (Appendix A). The divergence among psychrophilic strains seems more evident by the POCPs rather than ANI values.

### 3.6. Insertion Sequences and Restriction-Modification System Abundance

To further characterize the genomic structure of the genomes analyzed in this study, their insertion sequence (IS) and R-M system repertoires were described. Three main patterns of IS composition were found in the 28 genomes included in the analysis, as described elsewhere [28]. This pattern shows a trend indicating that the mesophilic, typical psychrophilic, and atypical psychrophilic strains include clear differences in their IS composition. The three genomes described in this study fall within the expected pattern of atypical psychrophilic strains, whereby the IS5 family is the most abundant. Notably, the ASA04 and ASA05 genomes show a higher number of insertion sequences compared to other atypical psychrophilic strains (Figure 6).

Furthermore, distinctive patterns of R-M coding sequences were found among the mesophilic, typical psychrophilic, and atypical psychrophilic strains (Figure 7). Remarkably, contrasting patterns appeared in the ISs and R-M coding sequences observed in the atypical psychrophilic strains, with respect to other strains. The IS number is more abundant in the atypical psychrophilic strains compared to the mesophilic strains. In contrast, the number of R-M coding sequences identified in the atypical psychrophilic strains is less abundant than in the mesophilic strains. Moreover, two distinctive patterns of R-M coding sequences are evident within the mesophilic strains, as well as among the atypical psychrophilic strains.

### 3.7. Virulence Factors

Virulence factors (VFs) were predicted in the genomes included in the present study and compared with complete *A. salmonicida* genomes available from NCBI [35]. The presence of virulence factor classes was mostly homogeneous between the compared strains (Appendix A). However, regarding exotoxin factors, the mesophilic strains showed the presence of actin cross-linking domain (ACD)-containing toxin factors, which were absent in the psychrophilic strains (Figure 8). Additionally, minor differences were observed regarding the pattern of exotoxin factors between the typical and atypical psychrophilic isolates. Specifically, the DNAse/genotoxin factor *clbF* was present in all typical psychrophilic but not in the atypical psychrophilic isolates, except for in samples J410, J411, and J409. Moreover, the pore-forming toxin factor *aerA*/*act* was present in most typical psychrophilic strains and in all mesophilic isolates yet was not in the atypical psychrophilic samples. Furthermore, the RNA N-glycosidase factor *stx2eA* highlights the differences within the atypical psychrophilic strains since it is present only in the S44, S121, S68, BR1900YR, RFAS1, AS1, AS2, and RZ6S-1 isolates. This virulence factor variability among atypical strains seems to be linked to the strains’ region of origin.

### 3.8. Characterization of Plasmids

All three genomes described in this study contained plasmids with distinct factor compositions (Figure 9). Genomes ASA04 and ASA05 contained two plasmids each, while genome CIBA_5017 contained one plasmid. However, the total number of unique plasmids was three since these were subsequently found to be redundant. Plasmid p01CIBA_5017 was found in the three strains reported in this study. A BLASTn search against the NCBI database showed that p01CIBA_5017 aligned with the *A. salmonicida* strain S44 plasmid pS44-3 (98% query coverage and 99.95% identity). In contrast, plasmid p02ASA04 was only found in ASA04, and after performing a BLASTn search against the NCBI database, this plasmid produced hits illustrating a maximum query coverage of 53%, which suggests that it has not been described previously. Finally, plasmid p01ASA05 did not include virulence or antibiotic resistance factors and was only found in the ASA05 strain (not shown).

### 3.9. Antibiotic Resistance Factors

The three genomes described in this study contain factors associated with antibiotic resistance factors (ARs) effective against several drug classes, including peptides, fluoroquinolones, diaminopyrimidines, cephalosporines, and carbapenems (Figure 10). Notably, analyzing the ARs present in plasmids from Chilean strains shows that the p02ASA04 plasmid also includes the *tet(A)* and *floR* factors with high similarity percentages (Figure 9). However, among these two factors, only *floR* was identified with a subject sequence coverage ≥ 80%. These factors are associated with resistance against the tetracycline and phenicol drug classes, respectively, which are commonly used in the Chilean aquaculture industry.

## 4. Discussion

Atypical furunculosis represents an endemic bacterial disease with significant morbidity and mortality which has traditionally been controlled through vaccination strategies. Notably, a substantial increase in the incidence of this disease was observed between 2022 and 2023, with a widespread geographical distribution in various freshwater production systems. This clinical and histopathological manifestation of cases is consistent with patterns described for furunculosis observed in salmonid fish. The histopathological findings are characterized by the presence of basophilic coccobacillary bacterial colonies in tissues, with inflammatory reactions being scant or absent. Other specific histological findings can be identified, such as cardiac inflammation, and hepatic and pancreatic necrosis, which are associated with co-infections with PRV and IPNV.

The genomic analysis of re-emergent *A. salmonicida* outbreaks in Atlantic salmon (*S. salar*) has provided significant insights into the pathogenesis and genomic diversity of this bacterial pathogen. The identified distinctive genomic patterns provide a comprehensive understanding of the complex dynamics associated with furunculosis, emphasizing its relevance in both salmonid and non-salmonid fish species across diverse aquatic environments. This study comprehensively examined clinical cases of atypical furunculosis in Atlantic salmon (*S. salar*) aquaculture in Chile, offering valuable insights into the epidemiological and clinical aspects of this disease. The selection of bacterial isolates from clinically diseased fish, characterized by the highest frequency of clinical signs, enabled a detailed genomic investigation and shed light on the pathogenesis and antibiotic resistance profiles of this bacterium in Chile. The gross pathology and histopathological findings corroborate previous observations [17], which noted a range of clinical signs, including periorbital hemorrhages, abdominal hemorrhages, and the presence of furuncles in affected fish. The histopathological analysis underscores extensive tissue colonization by *A. salmonicida*, primarily characterized by bacterial colonies that replace tissues with limited inflammatory responses. The prevalence of necrotic lesions in the epidermis and dermis is a key pathological feature which aligns with the observed clinical manifestations [36,68].

A pivotal contribution of this study is the genomic exploration of *A. salmonicida*, particularly in the distinctions between the mesophilic and psychrophilic strains, to confirm and extend recent findings in the field [26,27,28]. The phylogenomic analysis revealed three distinct clusters, typical psychrophilic, atypical psychrophilic, and mesophilic, which demonstrates the genetic diversity within this species and validates previous observations [27,28]. The strains investigated in this study are classified within the atypical psychrophilic cluster. Five subspecies of *A. salmonicida* have been described; however, the assignment of different isolates to each of these subspecies has been debated, primarily through the comparative analysis of complete genomes. Such analyses have revealed that the subspecies classification does not reflect the divergence at the core genome level [27]. In contrast to the classification into typical psychrophilic, atypical psychrophilic, and mesophilic groups, the last appears to have a genomic basis that is evident at the phylogenetic level and in the distribution of ISs and coding sequences in the R-M system. Notably, there is a higher abundance of ISs among atypical psychrophilic strains, confirming previous studies [26,27,28]. Also, the presence of specific GEIs could serve as geographical origin indicators [21,22,23,24]. However, none of the six described GEIs were found in the Chilean strains reported in this study.

The unique gene families found in these strains highlight the genomic complexity within atypical psychrophilic *A. salmonicida* and further emphasize the need for tailored genomic characterization. Thus, the analysis of virulence factors is of paramount importance. Specifically, since variations in exotoxin factors, particularly the actin cross-linking domain (ACD)-containing toxins, are present in the mesophilic strains but not in the psychrophilic strains, the potential role of these factors in pathogenicity is underscored. Distinctions in the presence of virulence genes such as *clbF* and *aerA*/*act* further accentuate the genetic diversity within typical and atypical psychrophilic strains. The selective presence of the RNA N-glycosidase factor *stx2eA* among atypical psychrophilic strains serves as an essential marker for distinguishing between subgroups within this classification. Plasmid characterization reveals differences in plasmid content, emphasizing the contribution of each plasmid to virulence and genome plasticity. However, small plasmids might have been missing due to the long-read sampling approach. The presence of antibiotic resistance factors within these genomes raises concerns regarding the development of antibiotic resistance and necessitates careful management strategies in Chilean aquaculture settings.

Furthermore, the diverse patterns in the insertion sequences and restriction-modification system coding sequences underscore the genomic structural differences among strains. REases function as regulatory agents, limiting the entry of foreign DNA with non-native methylation patterns to modulate genetic influx. This impediment in effect may also play a role in preserving bacterial species. Corroborating this, many *E. coli* and *Salmonella enterica* serovar Typhimurium strains possess specific genomic loci, referred to as “immigration control regions”, which are abundant in R-M systems [69]. Therefore, R-M systems could achieve greater genetic isolation by regulating the uptake of environmental DNA. This would allow distinctive R-M patterns to be established among closely related strains, further genetically isolating them. As several genetic variants distinctly accumulate, the related bacterial strains gradually evolve and become more distant, until eventually becoming different species entirely. Additionally, studies on type III-like R-M enzymes have uncovered the function of these REases, which acted as a significant obstacle to the HGT in clinical variants of methicillin-resistant *S. aureus* [70]. The distinctive R-M coding sequence patterns observed among typical psychrophilic, atypical psychrophilic, and mesophilic *A. salmonicida* strains highlight their genomic diversity. Further, two distinctive patterns of R-M coding sequences are observed within the atypical psychrophilic and mesophilic strains. However, the impact that the R-M system may have on the genomic diversity of *A. salmonicida* requires further investigation.

## Figures and Tables

**Figure 1 microorganisms-12-00064-f001:**
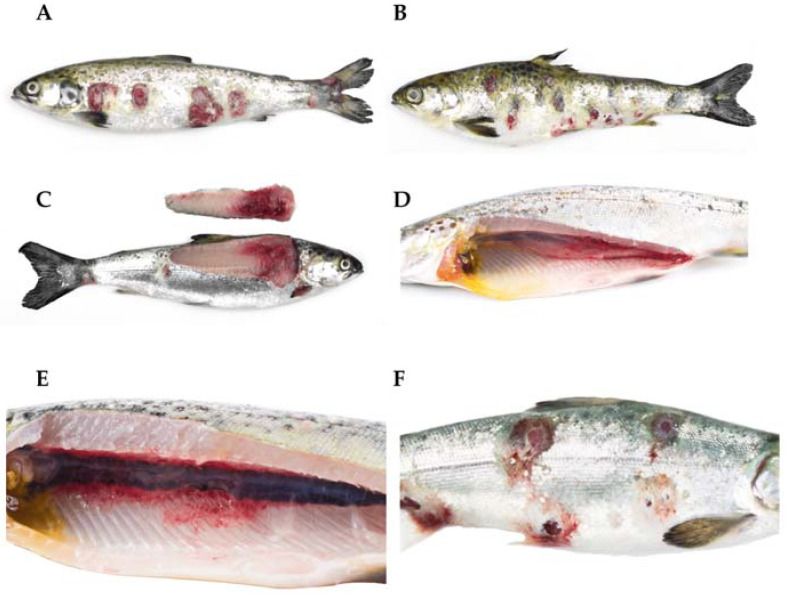
Atlantic salmon (*Salmo salar*) in the freshwater phase are affected by a clinical presentation of atypical furunculosis caused by atypical *Aeromonas salmonicida*. (**A**,**B**) Multiple ulcers are observed, with some surrounded by a white-colored border. (**C**) Loss of structure and muscle hemorrhages are observed. (**D**) Petechial hemorrhages are observed in the swim bladder. (**E**) Atlantic salmon (*S. salar*) affected by a clinical picture of atypical furunculosis (atypical *A. salmonicida*). The presence of perirenal muscle hemorrhages is observed. (**F**) Atlantic salmon (*S. salar*) affected by a clinical picture of atypical furunculosis (atypical *A. salmonicida*). Periocular hemorrhage, exophthalmia, and multiple ulcers are noted, some of which have coalesced.

**Figure 2 microorganisms-12-00064-f002:**
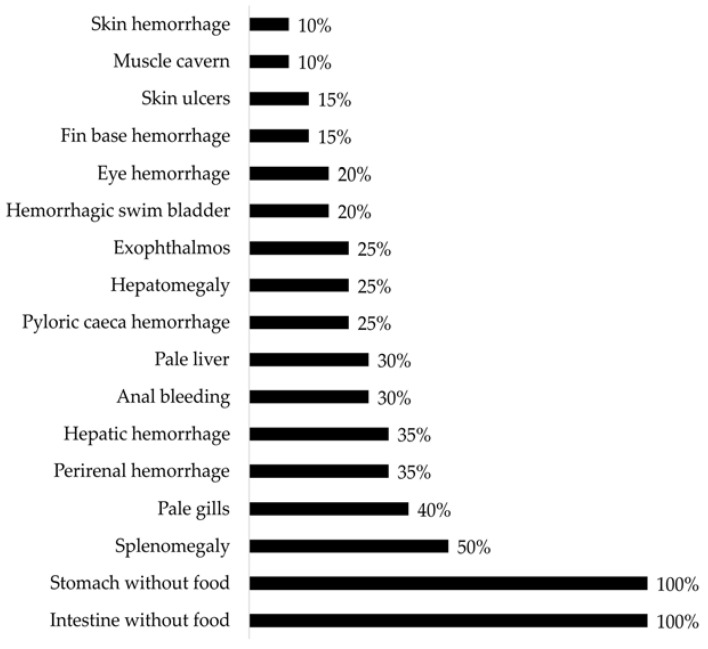
Gross pathology findings were observed in Atlantic salmon (*Salmo salar*) presenting clinical cases of atypical furunculosis between 2022 and 2023, in a total sample size (*n*) of 294.

**Figure 3 microorganisms-12-00064-f003:**
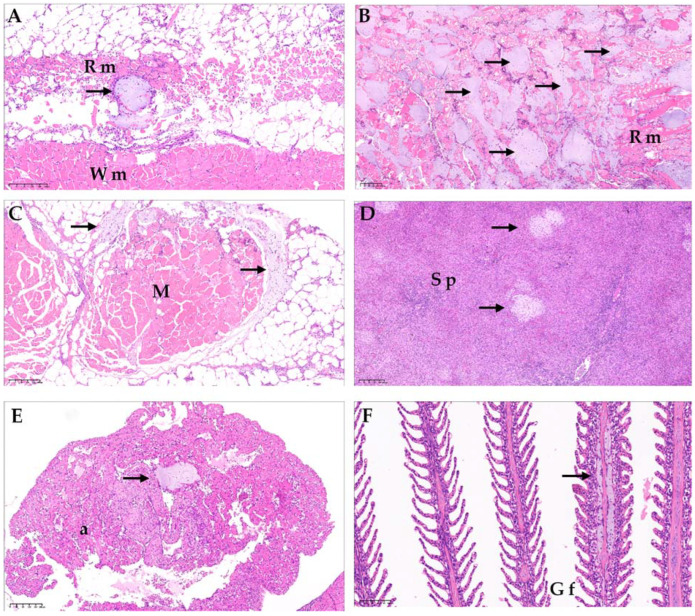
(**A**) Atlantic salmon (*Salmo salar*), red muscle (H&E). Colonies of bacteria with coccobacillary morphology and grey colored are observed in red muscle (Rm), consistent with atypical *Aeromonas salmonicida* infection (arrow). (**B**) Atlantic salmon (*S. salar*), red muscle (Rm) (H&E). Observation of diffused infection of bacteria with coccobacillary morphology, consistent with atypical *A. salmonicida* infection. Additionally, hemorrhage and necrosis of the muscle tissue are observed. (**C**) Atlantic salmon (*Salmo salar*), muscle tissue (H&E). Surrounding the muscular tissue (M), the presence of coccobacillary bacterial colonies consistent with infection by atypical *A. salmonicida* is observed (arrow). Additionally, muscular necrosis and hemorrhage are observed. (**D**) Atlantic salmon (*S. salar*), spleen (H&E). In the splenic parenchyma (Sp), multiple colonies of bacteria with coccobacillary morphology are observed, consistent with atypical *A. salmonicida* infection (arrows). (**E**) Atlantic salmon (*S. salar*) Auricle (H&E). Colonies of bacteria with coccobacillary morphology are observed in the auricle (a), consistent with atypical *A. salmonicida* infection (arrow). (**F**) Atlantic salmon (*S. salar*), gill (H&E). Bacterial infiltration of the center gill filament (Gf) with coccobacillary morphology is observed, consistent with atypical *A. salmonicida* infection (arrow). Wm: white muscle.

**Figure 4 microorganisms-12-00064-f004:**
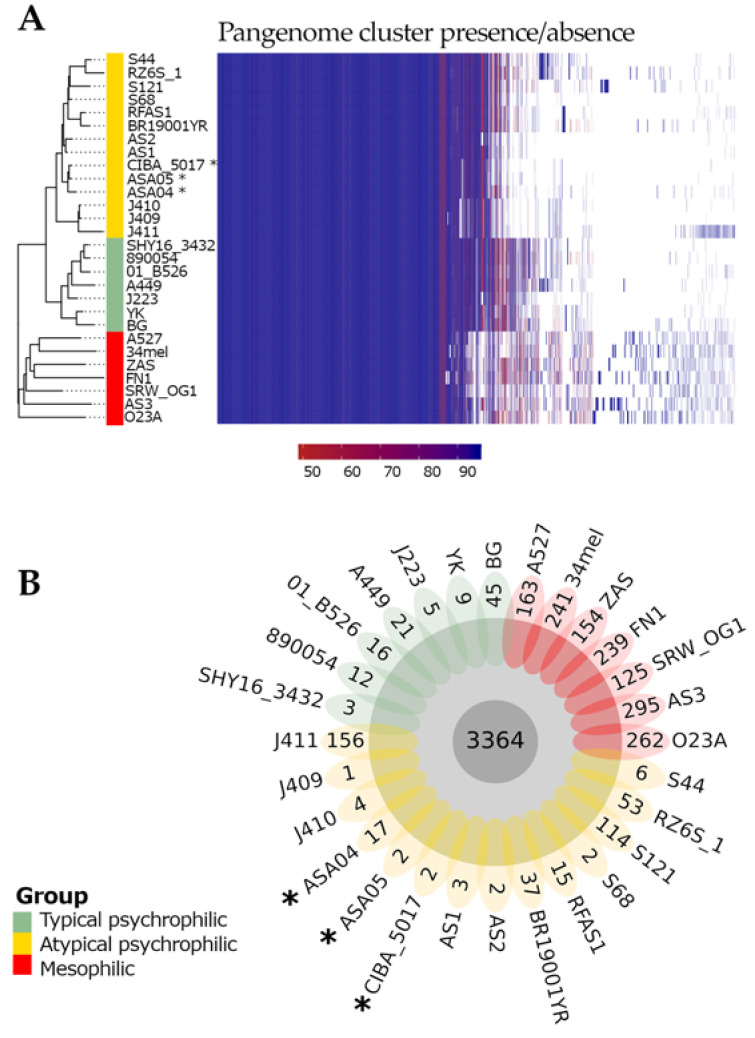
Pan-genome analysis of *Aeromonas salmonicida* strains. (**A**) Shared gene presence per isolate, ordered alongside the phylogenetic tree. Gene family presence is indicated by colored blocks per column, reflecting the corresponding percentage identity thresholds. (**B**) Core and unique gene families per genome. *: Genomes described in this study.

**Figure 5 microorganisms-12-00064-f005:**
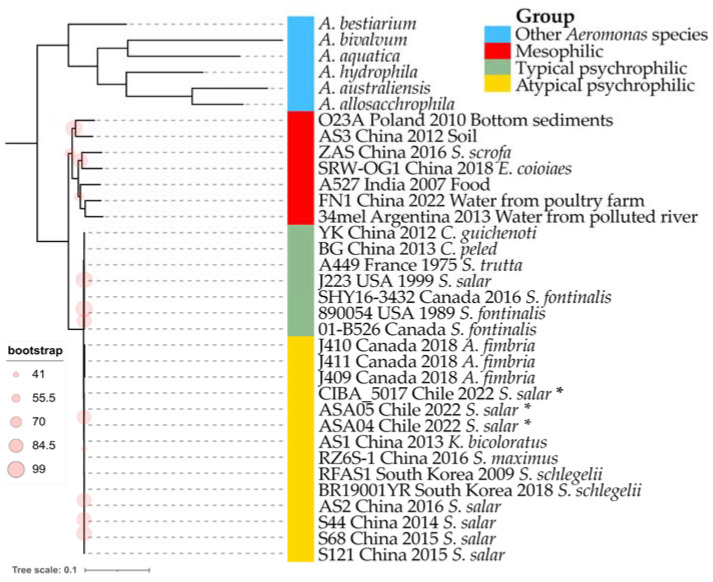
Phylogenomic analysis of *Aeromonas salmonicida* strains. Maximum likelihood phylogenetic tree constructed from a core genome alignment encompassing 89,915 SNPs. * Genomes described in this study. Bootstrap frequencies above 70% are represented as red circles. Other representative *Aeromonas* species are included as outgroup isolates. The country of origin, year of collection, and host/source are shown if available.

**Figure 6 microorganisms-12-00064-f006:**
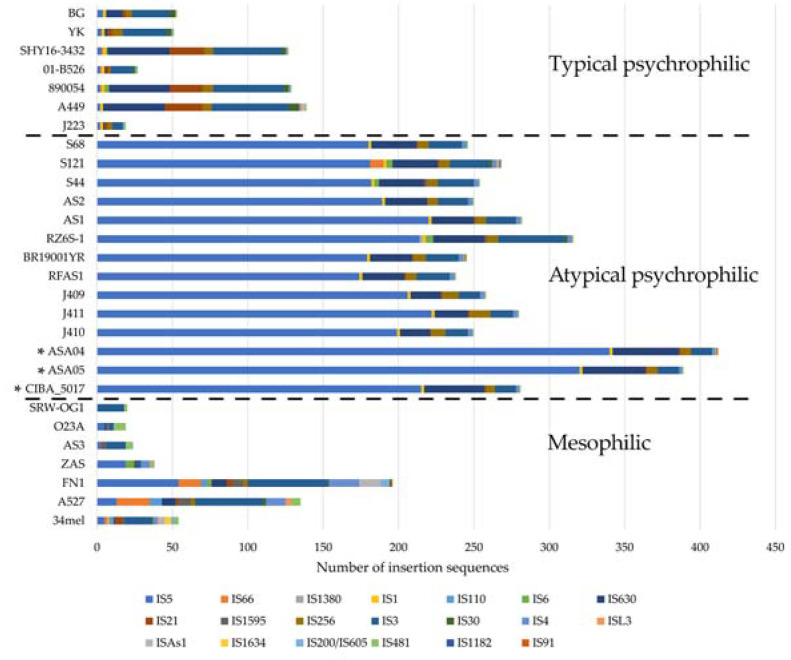
Distribution of insertion sequences (ISs) classified by family among *Aeromonas salmonicida* isolates. * Genomes described in this study.

**Figure 7 microorganisms-12-00064-f007:**
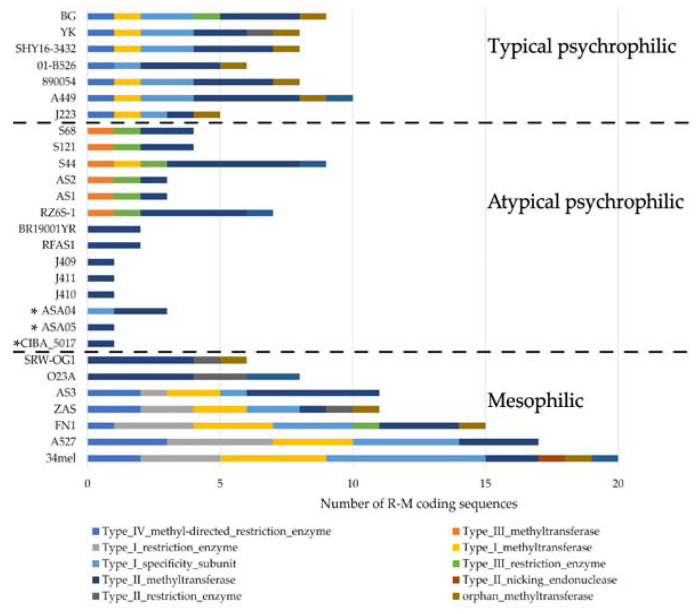
Distribution of restriction-modification (R-M) coding sequences among *Aeromonas salmonicida* isolates. Types of R-M coding sequences are shown in colors. *A. salmonicida* strains are grouped by typical psychrophilic, atypical psychrophilic, and mesophilic, and separated by a dotted line. * Genomes described in this study.

**Figure 8 microorganisms-12-00064-f008:**
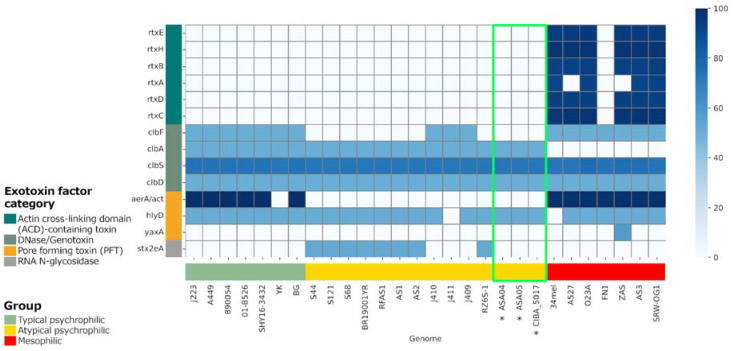
Profile of exotoxin virulence factors in complete *Aeromonas salmonicida* genomes. The presence and absence of exotoxin factors were predicted based on local alignments with the VFDB entries. The similarity percentage with the identified factors is shown. * Genomes reported in this study, with their results highlighted in the green box.

**Figure 9 microorganisms-12-00064-f009:**
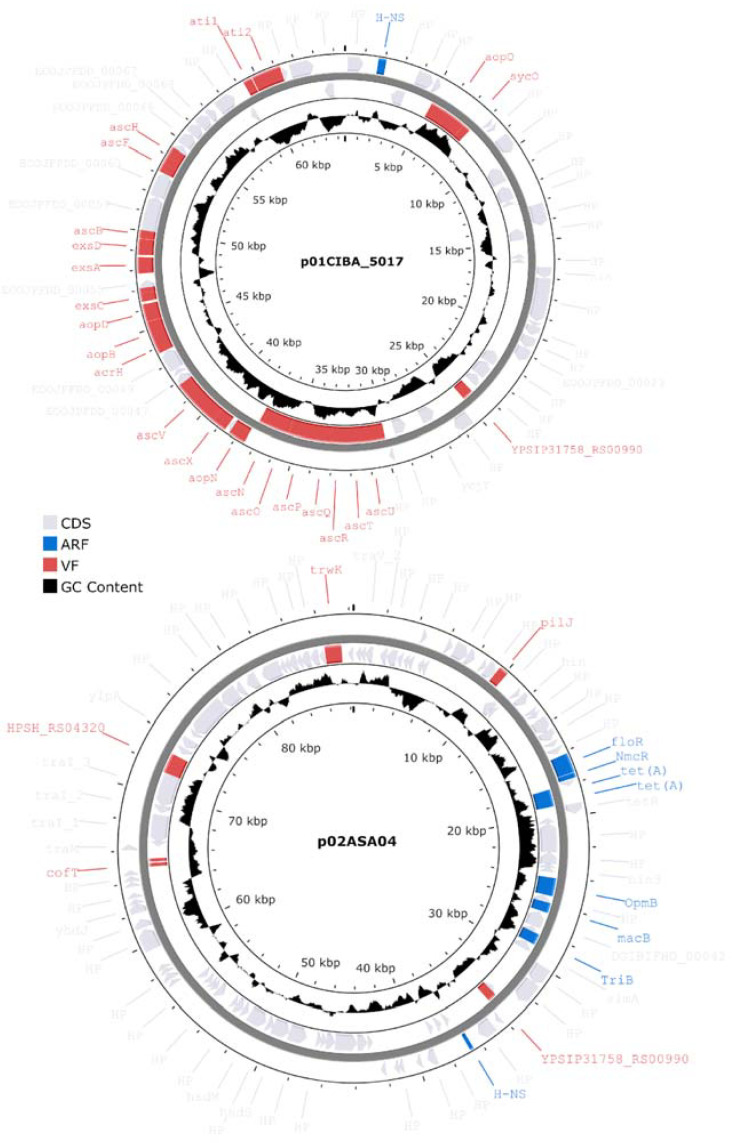
Virulence and antibiotic resistance factors identified in *Aeromonas salmonicida* plasmids. CDS: coding sequence, ARF: antibiotic resistance factor, VF: virulence factor.

**Figure 10 microorganisms-12-00064-f010:**
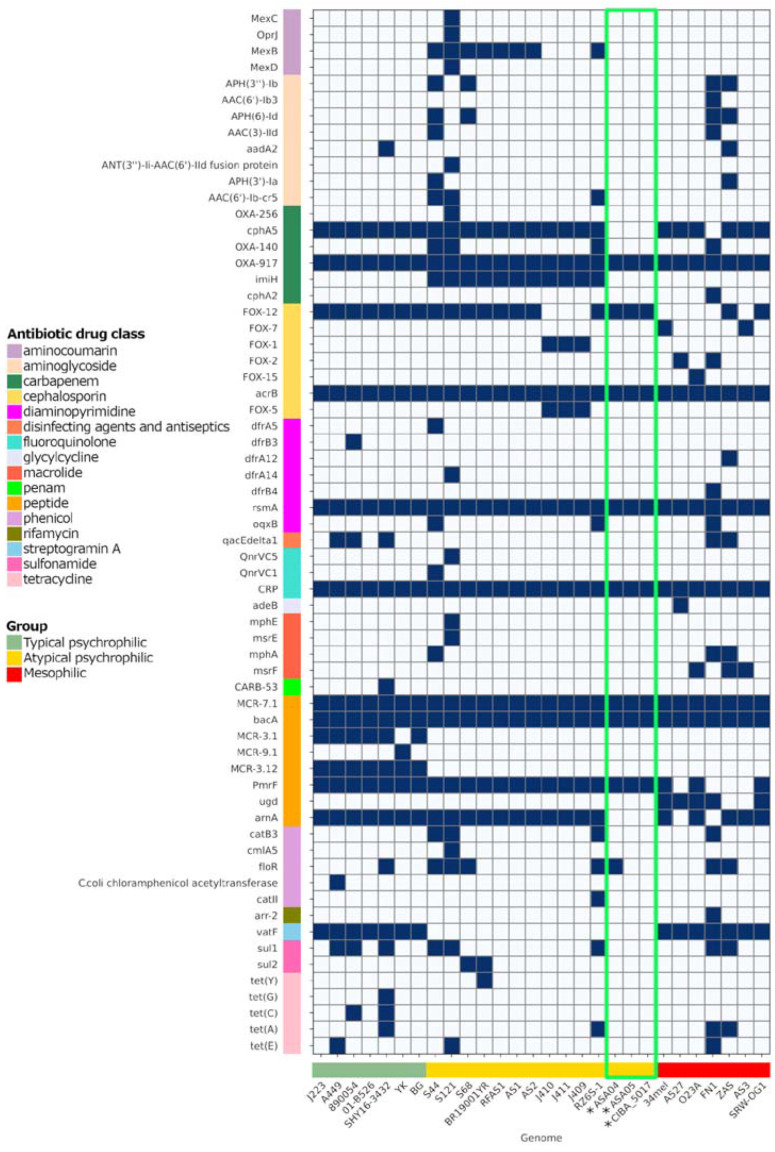
Antibiotic resistance repertoire in *Aeromonas salmonicida* complete genomes. The presence and absence of the factor are represented in blue and white, respectively. * Genomes reported in this study, with their results highlighted in the green box. Factors were filtered and considered as present as long as the local alignments had a similarity percentage ≥ 50%, e-value ≤ 1 × 10^−6^, and subject sequence coverage ≥ 80%.

**Table 1 microorganisms-12-00064-t001:** Epidemiological background and laboratory results for the cases analyzed in this study.

Case ID	4978	4988	5131	7003	7031	7081	7088	7124	7167	7175	7174	7215	7216	7214
Date	10-Mar	14-Mar	11-May	23-May	03-Jun	23-Jun	30-Jun	21-Jul	08-Aug	12-Aug	12-Aug	18-Aug	24-Aug	25-Aug
Region	X	X	X	X	X	XII	XII	XII	X	X	X	XII	XII	X
Salinity	FW	FW	BW	FW	FW	BW	BW	BW	BW	BW	FW	BW	BW	BW
Development stage	Pre-smolt	Pre-smolt	Pre-smolt	Pre-smolt	Pre-smolt	Pre-smolt	Pre-smolt	Pre-smolt	Pre-smolt	Pre-smolt	Pre-smolt	Pre-smolt	Pre-smolt	Pre-smolt
Fish sampled	20	7	10	30	6	7	7	10	10	11	24	1	1	1
Average ASA Ct	22.8	26.1	29.2	28.4	23.8	27.5	26.2	24.4	27.1	29.8	21.4	26.1	32.8	22.5
Coinfection	PRV	IPNV	IPNV	R. salm	PRV	IPNV-F.psy	IPNV-PRV	IPNV-PRV	-	-	IPNV-PRV	-	-	-
Bacterial isolation	Positive	Positive	Positive	Negative	Negative	Negative	Negative	Negative	Positive	Negative	Negative	Negative	Negative	Negative
GenBank accession	-	-	-	-	-	-	-	-	-	-	-	-	-	-
Case ID	7198	7213	7217	7385		7388	7387	7412	7416	7435	CIBA-5017		ASA04	ASA05
Date	26-Aug	26-Aug	06-Sept	20-Oct		21-Oct	21-Oct	27-Oct	28-Oct	04-Nov	25-Mar		22-Nov	23-Nov
Region	X	X	X	X		X	X	X	X	X	X		X	X
Salinity	FW	FW	FW	FW		BW	FW	FW	FW	BW	FW		FW	FW
Development stage	Pre-smolt	Pre-smolt	Pre-smolt	Pre-smolt		Pre-smolt	Pre-smolt	Pre-smolt	Pre-smolt	Pre-smolt	Smolt		Smolt	Smolt
Fish sampled	10	1	7	24		24	26	15	10	15	20		25	11
Average ASA Ct	29.6	26.2	23.2	30.6		26.6	24.3	30.8	29.1	27.0	28.8		29.7	26.1
Coinfection	IPNV	-	IPNV	-		-	-	-	PRV	PRV	-		IPNV-PRV	IPNV-PRV
Bacterial isolation	Positive	Negative	Negative	Negative		Negative	Positive	Negative	Negative	Negative	Positive		Positive	Positive
GenBank accession	-	-	-	-		-	-	-	-	-	CP139910		CP139915	CP139912

BW: brackishwater, FW: freshwater, IPNV: infectious pancreatic necrosis virus, PRV: piscine orthoreovirus, R.salm: *Renibacterium salmoninarum*, F.psy: *Flavobacterium psychrophilum*, ASA: *Aeromonas salmonicida*.

## Data Availability

The genome assemblies described in this study are openly available on NCBI, reference number PRJNA1045586.

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
