# Peer review of "Genomics of Re-Emergent Aeromonas salmonicida in Atlantic Salmon Outbreaks"

_microorganisms, 2023, doi:10.3390/microorganisms12010064_

Round 1
Reviewer 1 Report
Comments and Suggestions for Authors
1. Recalculating aquaculture technology is preferred over vaccination. Please sign it.
2. What are your observations about the presence of the detection of a novel genomic entity, denoted as AsaGEI1a. what about in other 4 members, and its physiological role need to me mention
3. Mentions of the most common antibiotics that were reported to be resistant to the strain
4. Kidney and spleen are selected for molecular biology work? Why not liver, as the authors mentioned the collection of other organ in bacterial isolation?
5. Piscine orthoreovirus (PRV), infectious salmon anemia virus (ISAV), and infectious pancreatic necrosis virus (IPNV) detection were also performed. Is there any significance to detecting these viruses?
6. How many strains were isolated, and what are the criteria for the selection of ASA04 and ASA05 strains for whole genome sequencing? Are these gene bank numbers? ...in table 1. The author mentions the gene-back accession number is not given. Under virulence factors J410, J411, and J409.?? Are described or clarified it.
7. Provide the primer sequence used in PCR.
8. 1: Some scientific patterns (itellic)need to be checked in the 2.3 methodology and introduction sections.
9. Furunculosis in salmonids: what are the basic characteristics of the disease? A line in the introduction needs to be added.
10. The sentence needs to be checked. The first report of A. salmonicida atypica isolates in Atlantic salmon (S. salar) cultured in Chilean seawater was.
Comments on the Quality of English Languageminor
Author Response
Dear Reviewer
Attached file with all answers.
Regards,

Reviewer 2 Report
Comments and Suggestions for Authors
This study explores the genomic details of re-emergent A. salmonicida outbreaks in Atlantic salmon, focusing on three Chilean isolates (ASA04, ASA05, and CIBA_5017). The analyses presented in the study are interesting. However, the manuscript contains many confusing elements in its introduction, potential errors in the experimental approach and important limitations in the data analysis which must be absolutely considered for the study to be complete, rigorous and relevant.
1. The absence of line numbers in the manuscript complicates the production of this evaluation. I hope that the authors will understand to which part of their text my comments refer. The comments with ** are of the utmost importance.
2. It should be added that the study concerns Chilean strains in the title of the article.
3. ** The authors must explain the meaning they give to the word re-emergence in their study and they must provide more details in their description of the samples to bring out this aspect of re-emergence.
4. ** First paragraph, reference #1: Since the publication of this reference, many things have changed regarding the A. salmonicida species. The first paragraph should be reconsidered to at least remove the sentence saying that the optimal growth is 22-25 while it is now well known that this temperature affects the majority of the typical strains leading to loss of virulence factors. See all the literature done since 80s and more particularly the last decade about that, especially about the vapA gene and the pAsa5 plasmid.
5. ** Introduction, “can be divided into five subspecies”: Yes, the species A. salmonicida actually includes 5 recognized subspecies, but there are also a very large number of strains, particularly among mesophiles, which are potentially part of new subspecies which are not yet defined. This should be presented in the introduction with the appropriate references.
6. Introduction, “These strains show optimal growth at temperatures ranging from 22 to 25 °C.”: Did the authors mean strains of subspecies masoucida, achromogenes, and smithia? In the case of subspecies salmonicida (typical), this is not the case.
7. ** Introduction, “Notably, among the known subspecies of A. salmonicida, only A. salmonicida pectinolytica has been observed growing at temperatures exceeding 35 °C.”: This is not true. Many articles since at least 2016 have described numerous mesophilic strains of A. salmonicida that can grow at 37 and even above. This information should be presented in the Introduction with appropriate references.
8. Introduction, second paragraph: “atypical” instead of “atypica”.
9. Introduction, second paragraph, first sentence: Rewrite this sentence.
10. Introduction, second paragraph, second sentence: It can be interesting to have a map showing the progression of the detection of the disease upon time in Chile.
11. Add a reference after “reduced disease prevalence”.
12. Add a reference after “among unvaccinated fish groups”.
13. The last two paragraphs of the introduction should be divided in many smaller ones: One paragraph per idea or study cited.
14. Some words are in italic like “atypical” and “mesophilic”, but it is not necessary.
15. Many words in the text should be in italics:
AsaGEI
IS (example: ISAs4)
Gene names
Names of microbes in the text and in the references
16 ** The authors should define what they consider as mesophilic A. salmonicida. In the literature it is defined as strains being able to grow at 37 C and even higher.
17. ** Section about AsaGEI in the introduction: This part of the text needs to be updated. Now, 6 AsaGEIs have been described. Two in North America (1a, 2a), three in Europe (1b, 2b and 2d) and one in China (1c). PubMed ID (PMID) of the missing references: 33605980, 27493011, 26048417.
18. “Recently, there is a noteworthy surge in ARGs these aquatic environmental strains.”: Reconsider the writing of this sentence.
19. **At the end of the introduction, provide more information about the Chilian context and clearer objective for this study.
20. M&M: Which vaccine? Provide more information.
21. M&M, “high mortality rates “: Were the symptoms caused by furunculosis?
22. M&M: Do the isolated strains produced pigment. I guess not, but I propose that you specify it.
23. M&M, section 2.3: At least indicate which gene or DNA region was targeted by the primers.
24. **M&M, section 2.4: Provide more information on this part. What were the colonial features considered? How was performed the isolation and pure culture procedure?
25. M&M, section 2.5: Why were the three strains not sequenced by the same approach?
26. **M&M, section 2.5: Please provide more information on the raw sequencing data statistics of all strains sequenced including coverage for PacBio.
27. “MinION Mk1C was converted into” instead of “MinION Mk1C were converted into”.
28. “Overall, 27 clinical cases of atypical furunculosis were reported in Atlantic salmon (S. salar) during 2022 and analyzed in this study.” Do the authors mean in Chile? If yes, specify it.
29. The legends of the figures and tables do not contain enough explanation to fully understand what is presented. In particular, the meaning of abbreviations and other details must be explained.
30. **There is information missing from Table 1:
-Explain all the abbreviations
-Bacterial isolation : Why three outcomes here? Some samples not tested?
-GenBank accession: In which case there are sequenced strains (the last row of the table is empty).
31. I am not an expert of fish pathology and histology. So I was not able to review sections 3.2 and 3.3.
32. **Figure 4: There is not enough information provided in the figure legend to fully understand the different analyzes presented by readers less familiar with this type of analysis. The authors need to better explain each of the graphic representations.
33. **Section 3.5: The authors should perform average nucleotide identities (ANIs) and percentage of orthologous common proteins (POCPs) analyses on genomic data that provide very helpful information when comparing genomes that are highly similar as it is the case here.
34. Figure 5: To simplify the representation of the tree, only indicate bootstrap values that are not 100.
35. **Supplementary figure1 is relevant to the study and should be included in the main manuscript.
36. ** Reference #17 (Vasquez et al.) was not the first study addressing the question of ISs in A. salmonicida genome. In 2016, Vincent et al. (PMID 26753691) did the same type of analysis. In this case, A. salmonicida ssp. smithia was also included with other atypical A. salmonicida strains. It is important to mention that a very large number of ISs was detected in A. salmonicida ssp. smithia genome as for the genomes analyzed in the revised study. The authors should include the Vincent et al. paper in their discussion.
37. ** Is it possible that the variability in the virulence factor genes is linked to the region of origin for the atypical strains or their subspecies? The POCP analysis will help to answer this question by allowing sub-grouping of the strains analyzed.
38. ** Figure 7: The vapA protein (of the A-layer) should also be included in this analysis even if it is not found on VFDB. Provide, in the legend of the figure, information about the virulence genes included in this figure.
39 ** Are there other plasmids present in these strains other than the plasmids described in this paragraph? Small plasmids are often found in strains of A. salmonicida and long-read sequencing methods may not be able to detect these small plasmids due to their particular characteristic versus large plasmids that behave like chromosomal DNA. So, it is possible that the authors may have missed these small mobile DNA elements. It is, for example, mentioned that "Plasmid p01CIBA_5017 was found in the ASA04, ASA05, and CIBA_5017 strains. This plasmid was identified as the A. salmonicida strain S44 plasmid pS44-3." The S44 strain is known to also have a small plasmid (pS44-5) which resembles the pAsal1 plasmid found in typical strains, but with one less ISAs11. Authors must demonstrate the presence or absence of small plasmids in the sequenced strains; otherwise, they may introduce errors into the databases and into their study. See pS44-5 plasmid: NZ_CP022180.1. See more information about S44 strain plasmids in the supplementary table of the article PMID 33040386. The authors can use the plasmid profile experiment by electrophoresis as described by Boyd et al. (PMID 12932739) to visualize small plasmids. If they find them in the strains, they will need to perform short-read sequencing to have their sequence.
40. ** Figure 8: More details are needed in the legend to understand the maps.
41. Section 3.9, “ASA04p2”: Use the same name as on the map (p02ASA04).
41. ** Why some of the ARGs shown on the maps are not in the list of the table in Figure 9? For example, the tet(A) gene is not shown for the strain bearing the p02ASA04 on figure 9. To be corrected.
42. ** What about AsaGEI in the strains analyzed? The AsaGEIs are mentioned in the introduction of the article but not mentioned in the analysis. Partial sequences of AsaGEI elements seem to be present in some atypical strains based on a rapid analysis I made on NCBI (wgs database). Even if AsaGEIs are not found in the genome of the analyzed strains, it should be mentioned.
43. ** While the discussion is concise, it lacks detailed coverage of certain study aspects. Notably, there is no mention of AsaGEIs in the analyzed strains, and it does not establish a connection between the examined strains and subspecies previously documented in the literature. Addressing these points would enhance the comprehensiveness of the discussion.
44. ** References are needed at least here in the Discussion:
-After “an outbreak duration of 8 weeks”
-After “with the observed clinical manifestations”
-After “recent findings in the field”
-After “coding sequences in the R-M system”
44. “does not reflect the divergence” instead of “does not reflect divergence”.
45. ** “The selective presence of the RNA N-glycosidase factor stx2eA among atypical psychrophilic strains serves as an essential marker for distinguishing between subgroups within this classifica- tion.”: POCP analysis should also help to make this subgrouping probably based on official subspecies or putative subspecies.
46. Reference 13 is incomplete. Is it a PhD thesis? If yes, indicate it and from which university.
Comments on the Quality of English LanguageModerate editing of English language is required.
Author Response

(The authors gave the same response as above.)

Reviewer 3 Report
Comments and Suggestions for Authors
This is an interesting study on genome-encoded features of Aeromonas salmonicida, a bacterium that poses a significant threat to both salmonid and non-salmonid fish in diverse aquatic environments. The manuscript is clearly written and well organized.
The provided pdf file lacks line numeration, making it challenging to refer to specific part of the text. I acknowledge the inconvenience this poses to both you as the reader and me as the reviewer.
In the text, the term 'psychrophilic' is used to describe strains with an optimum temperature below 20°C. It is definitely more precise to use 'psychrotolerant' if these strains exhibit optimal growth between 22–25°C. Please explain or replace ‘psychrophilic’ with ‘psychrotolerant’.
Introduction section.
“Moreover, the establishment and implementation of a vapA-centric typing system facilitated the subtyping of 675 isolates of A. salmonicida.” Could you please elaborate on what the vapA-centric typing system means?
Methods section
2.5. Whole Genome Sequencing (WGS) and Bioinformatic Analysis
Which Guppy basecalling model was employed, and did you utilize GPU or CPU for the basecalling process? Additionally, what version of the Guppy software was used?
Genomes of ASA04 and ASA05 strains were assembled using Flye and Canu without subsequent quality check. Why?
Genome of CIBA_5017 strain was assembled using Unicycler. Was there any specific reason to use three different genome assemblers? Does this introduce any potential bias into subsequent genome analysis?
What criteria were used to search for gene families? Please, provide a definition for the term 'gene family'. What thresholds were utilized to identify othologous gene clusters (coverage, identity, e-value)?.
Could you refer to the methodology used for generating the heatmaps?
Results section
Table 1. Could you provide explanations for the abbreviations used?
3.4. Pan-Genome Characterization
“The genomes described in this study, ASA04, ASA05, and CIBA_5017, contained 17, 2, and 2 unique gene families, respectively.” This is very unusual to detect only a few gene families. Could you provide examples of similar results from other publications in the Discussion section?
Please describe results presented in Figure 4 in more detail. What is the biological meaning of figures within panels A, B, C, D? Please elaborate.
3.8. Characterization of Plasmids
“In contrast, plasmid p02ASA04 was only found in ASA04, and after performing a BLASTn search against the NCBI data-base, this plasmid produced hits illustrating a maximum query coverage of 53%, which suggests that it has not been described previously.” What is the blastn coverage for other plasmids in studied strains? How does low coverage explain the fact that the plasmid was not described?
Discussion
“Five subspecies of Aeromonas salmonicida have been described; however, the assignment of different isolates to each of these subspecies has been debated, primarily through a comparative analysis of the complete genomes.” Authors do not provide data on pairwise ANI values between the genomes of new isolates and characterized Aeromonas strains. Please classify your isolates according to https://doi.org/10.1099/ijsem.0.002516
Author Response

(The authors gave the same response as above.)
